# Exonucleases: Degrading DNA to Deal with Genome Damage, Cell Death, Inflammation and Cancer

**DOI:** 10.3390/cells11142157

**Published:** 2022-07-09

**Authors:** Joan Manils, Laura Marruecos, Concepció Soler

**Affiliations:** 1Serra Húnter Programme, Immunology Unit, Department of Pathology and Experimental Therapy, School of Medicine, Universitat de Barcelona, Feixa Llarga s/n, 08907 L’Hospitalet de Llobregat, Spain; joanmanils@ub.edu; 2Immunity, Inflammation and Cancer Group, Oncobell Program, Institut d’Investigació Biomèdica de Bellvitge—IDIBELL, 08907 L’Hospitalet de Llobregat, Spain; 3Breast Cancer Laboratory, Cancer Biology and Stem Cells Division, The Walter and Eliza Hall Institute of Medical Research, Parkville, VIC 3052, Australia; marruecos.l@wehi.edu.au; 4Immunology Unit, Department of Pathology and Experimental Therapy, School of Medicine, Universitat de Barcelona, 08007 Barcelona, Spain

**Keywords:** cancer, exonuclease, DNA repair, DNA degradation, inflammation, apoptosis

## Abstract

Although DNA degradation might seem an unwanted event, it is essential in many cellular processes that are key to maintaining genomic stability and cell and organism homeostasis. The capacity to cut out nucleotides one at a time from the end of a DNA chain is present in enzymes called exonucleases. Exonuclease activity might come from enzymes with multiple other functions or specialized enzymes only dedicated to this function. Exonucleases are involved in central pathways of cell biology such as DNA replication, repair, and death, as well as tuning the immune response. Of note, malfunctioning of these enzymes is associated with immune disorders and cancer. In this review, we will dissect the impact of DNA degradation on the DNA damage response and its links with inflammation and cancer.

## 1. Role of Exonucleases

The description by Watson and Crick of the structure of DNA in the early 1950s [1] led to a revolution in molecular biology. The capacity of DNA to store and replicate the information required for cells and organisms to live was later discovered. Nowadays, everyone knows that DNA is the essential genetic material containing the map and instructions of who we are. DNA is so important that eukaryotic cells dedicate a whole lipidic fence (nuclear envelope) and heavy compaction to protect it. It might be difficult to understand then, why a cell would want to degrade these precious nucleic acids. However, cells have hundreds of different proteins with the capacity for cutting nucleic acids, and such an investment in this activity indicates that eliminating DNA is vital.

DNA is made of two chains of polynucleotides. The building bricks of DNA, the nucleotides, contain three components, a sugar attached to a base containing nitrogen (adenine (A), thymine (T), guanine (G), or cytosine (C)) and a phosphate group that, through phosphodiester bonds, interlinks the 5′-phosphate end of one sugar to the 3′-hydroxyl end of the next sugar, forming the polynucleotide chains. Phosphodiester (P-O) bonds are among the most versatile and stable biochemical bridges between biomolecules [2]. However, nucleases are able to cleave one of the two phosphodiester bonds that link adjacent sugars. There are multiple types with multiple functions, but grossly one can divide nucleases according to the type of substrate they cleave (RNAses [3] or DNAses [4]) and wherein the nucleic acid chain they perform the cut (endo- or exonucleases). While endonucleases cut the P-O bond from inside the polynucleotide chain generating two oligonucleotides and can be sequence- or structure-specific [5], exonucleases hydrolyze the bonds from the outer ends of the chain. Exonucleases can sequentially cleave P-O bonds from 3′-OH or from 5′-P of a single or double DNA chain in a nonspecific manner, generating individual nucleotide monophosphates [6].

The molecular event of a chemical modification of the DNA structure triggers signalling cascades that ultimately produce a cellular response. To maintain genome integrity, cells have a DNA damage response (DDR) mechanism, a multiple pathway response that integrates DNA damage sensing, DNA repair machinery, halting of the cell cycle and if repair is not possible, cell death [7]. Lesions in DNA are sensed by specialized proteins such as ATM, DNA-PK, and ATR [8], which act depending on the type of lesion and the cell cycle phase. While the different factors required to repair a specific DNA lesion are being activated and recruited to the damaged sites, p53 protein receives the signals to stop the cell cycle [9], thus preventing the transmission of DNA lesions to the daughter cell. Exonuclease activity is important in all steps in this process, from DNA sensing and repair to cell death.

Distinct exonucleases, such as APE1 [10], EXO1 [11], FAN1 [12], and FEN1 [13], are important components of several DNA repair pathways, including base excision repair (BER), nucleotide excision repair (NER), mismatch repair (MMR), non-homologous end joining (NHEJ), homologous recombination (HR), single-strand break repair (SSBR), post-replication repair (PRR) or DNA damage tolerance (DDT), interstrand cross-link repair (ICL), stalled replication fork and hairpin structure repair, as well as polymerase proofreading, as detailed below. Their ability to cleave DNA allows the elimination of damaged or mismatched nucleotides, which facilitates subsequent insertion of the correct base [14].

Some other exonucleases take part in apoptosis. Apoptosis occurs in normal development, cell turnover, and lymphocyte maturation but also in response to stress such as infection or excessive DNA damage. During apoptosis, DNA is condensed and fragmented [15] to facilitate digestion by engulfing macrophages [16]. For instance, the apoptosis enhancing nuclease (AEN), an exonuclease [17] transcribed by activated p53, is required for p53-induced apoptosis [18]. TREX1 expression increases upon genotoxic damage [19] and contributes to cell death induced by GzmA [20]. GzmA is part of the SET complex, which is released by cytotoxic cells to degrade DNA, prevent its repair and ensure death [21]. Similarly, the keratinocyte-specific TREX2 exonuclease promotes the passage of UVB-irradiated keratinocytes to late non-reversible apoptotic stages [22]. Other exonucleases participate in the degradation of DNA upon apoptosis activation, such as ARTEMIS [23], FEN1 [13] and APE1 [10].

Foreign and self-nucleic acids pose a threat to the organism, and exonucleases play an important role in tuning the innate immune response. By degrading DNA from pathogens, exonucleases control both invader infection and type I interferon (IFN) immune responses that are driven by DNA-sensing proteins [24]. Because nucleic acid sensors can also recognize endogenous DNA [25], nucleases are pivotal in removing excessive endogenous DNA to prevent detection. Exonucleases like TREX1 in the cytosol and PLD3 and PLD4 in the endolysosomes regulate cytosolic cGAS/STING activation and endosomal TLR nucleic acid-sensing, respectively [26,27], thereby limiting DNA-driven autoimmune diseases, such as rheumatoid arthritis and lupus [28,29]. Of note, autoimmunity may also be a risk factor for cancer [30,31].

Hence, exonuclease activity might come from proteins with single or multiple functional domains, such as apoptotic nucleases and DNA polymerases respectively. As stated above, nucleotide cleavage by exonucleases is important in many and quite different cell processes, from DNA synthesis/repair to DNA degradation during cell death, including DNA-driven inflammatory responses, maintaining genome stability, and ensuring the viability of the organism (Figure 1) [32,33,34].

Here, we focus on proteins with robust exonuclease activity and their role in the DDR and cancer. Thus, we comment on AEN, APE1, ARTEMIS (DCLRE1C), EXD2, EXO1, EXOG, FAN1, FEN1, MRE11A, p53, PLD3, PDL4, POLD1, POLE, RAD9A, TREX1, TREX2, and WRN, most of them included in the recently curated list of DNA Damage Repair genes in cancer [35].

To ascertain functional interactions, we performed an analysis of the 18 above-mentioned exonucleases using the STRING database of known and predicted protein–protein interactions (Figure 2). Ten exonucleases (EXO1, WRN, p53, MRE11, RAD9A, DCLRE1C, FEN1, APEX1, POLE y POLD1) were interconnected, indicating that interactions between them have been described at least in curated databases, experiments or in the literature, and functionally associated. All these exonucleases were significantly associated with the general GO process “DNA metabolic process” (dark blue) and most of them participate in DNA repair pathways.

## 2. Exonucleases and Cancer

### 2.1. AEN

Apoptosis enhancing nuclease (AEN), also known as ISG20L1, is an exonuclease that is highly efficient at processing 3′ DNA ends [17]. It is transcribed by activated p53 and promotes both single- and double-stranded DNA and RNA digestion to amplify apoptosis. If absent, cells are resistant to this type of cell death [18]. Importantly, expression of AEN not only promotes but is also required for autophagy [36].

AEN expression is upregulated in human peripheral blood mononuclear cells upon low-energy X-ray exposure during dual-energy computed tomography (DECT) [37] and cyclophosphamide treatment, stimulating the proinflammatory cell death of both tumour and blood cells and thus enhancing the efficacy of immunotherapy [38]. Moreover, bufalin also induced the expression of AEN in lung cancer cells in vitro [39]. AEN was included in a marker signature that can identify patients with a high risk of biochemical recurrence in prostate cancer (Table 1). High levels of gene expression, together with other genes, can predict recurrence [40]. Similarly, enhanced expression of AEN was used as a prognostic marker in an RNA-binding protein signature for colorectal cancer [41]. Given its response to genotoxins and its links to p53 and apoptosis, it is rather surprising that there is little existing knowledge on the role of AEN in cancer.

### 2.2. APE1

Apurinic/apyrimidinic endonuclease 1 (APE1), APEX1 or reduction-oxidation factor (Ref1), is a multifunctional enzyme, its main function being to incise the phosphodiester bond immediately 5′ to apurinic/apyrimidinic (AP) sites to generate single-strand breaks (SSBs). However, this protein also possesses 3′–5′ exonuclease activity [10]. It is involved in maintaining genome stability, participating in several DNA repair pathways such as tri-nucleotide repair (TNR) by the removal of hairpin structures and BER, digestion of matched and mismatched 3′ ends of duplex DNA structures, and the recognition of SSBs to induce their repair, and in apoptosis by exonucleolytic digestion of chromosomal fragments. It also prevents the formation of double-strand breaks (DSBs) during the repair of bi-stranded clustered DNA damage by nucleotide incision repair (NIR), which repairs oxidative damage in nucleotides, and interacts with POLB [10] to carry out proofreading.

Although as an endonuclease it is highly specific for AP sites, as an exonuclease it can recognize a wide range of abnormal nucleotides that are generated by oxidative stress, ionizing radiation (IR), or drug treatments [42], such as anti-cancer and anti-viral therapies. Therefore, inhibiting APE1 is an attractive approach for killing tumour cells; in fact, some APE1 inhibitors are already in clinical trials [43,44,45]. Reducing the levels of APE1 sensitizes the cells to PARP inhibitor treatment [46], hence combined therapy with PARP and APE1 inhibitors has been suggested to be highly effective in breast cancer.

Cell lines deficient for APE1 accumulate DNA damage and induce stress response pathways such as senescence [47,48,49]. *Ape1* knockout mice (Table 2) showed embryonic lethality [50]. However, conditional deletion of the gene early after birth induced impaired growth, reduced organ size, and increased cellular senescence in tissues like skin or colon [49]. These mice also showed an accumulation of replication-blocking lesions with increased DDR foci at telomeres, which are known to accumulate high levels of oxidative damage [51]. Hemizygous mice showed normal life expectancy but higher spontaneous mutations and elevated risk of tumorigenesis, including lymphomas, sarcomas, and adenocarcinomas [52,53,54].

In several cancers (including lung, colorectal, cervical, prostate, bladder, gastric, hepatic, glioblastoma, osteosarcoma, head and neck, ovarian, and breast) high APE1 expression or aberrant cytoplasmic distribution (Table 1) have been associated with tumour aggressiveness, poor prognosis or increased resistance to DNA-damaging agents [55,56]. For instance, in breast cancer, high APE1 expression has been reported in tumor-initiating cells [57], potentially protecting these cells from irradiation-induced oxidative stress and consequent senescence. On the other hand, the presence of cells with low/deficient APE1 expression may be linked to a good prognosis because this increases senescence, which acts as a tumour suppressor. Some somatic mutations have also been found in glioblastoma [58] and endometrial cancer [59], including the R237C substitution, which leads to reduced exonuclease activity [60]. Besides, some polymorphisms in the APE1 promoter have been associated with a decreased risk of lung cancer [61,62].

### 2.3. ARTEMIS

ARTEMIS, also known as SNM1C/DCLRE1C, is a member of the metallo-b-lactamase superfamily, characterized by their ability to hydrolyze DNA or RNA. ARTEMIS is essential for the NHEJ pathway, where it removes 5′ and 3′-overhangs to join duplex DNA ends or hairpin openings [63,64]. It also facilitates early site-specific chromosome breakage during apoptosis [23]. Although its main nuclease activity acts in a protein kinase C (PKC)-dependent manner, its 5–3′ exonuclease activity is independent of PKC and allows it to function more efficiently in 1- or 2-nucleotide 5′ overhangs, which are too short for endonucleolytic activity and occur following exposure to IR [63,64]. Hence, cells or patients lacking ARTEMIS cannot repair damage caused by IR [65,66] or alkylating agents used in chemotherapy [67]. Moreover, patients with deficiency or mutations (frequently found among Native Americans [68,69]) in ARTEMIS suffer from severe combined immunodeficiency (SCID) (T^−^B^−^NK^+^) [66] owing to the importance of NHEJ during B and T lymphocyte maturation, where V(D)J recombination is initiated by the creation of DSBs. ARTEMIS null mice also present SCID (Table 2) but they do not exhibit a higher risk of cancer [70]. However, when combined with *Trp53* loss, accelerated tumorigenesis has been observed. These mice develop especially aggressive B-cell lymphomas, indicating that ARTEMIS acts as a tumour suppressor in the absence of p53 [71]. Therefore, the defective function of ARTEMIS leads to unrepaired DSBs and malignant transformation of cells that escape apoptosis. ARTEMIS can also act as a negative regulator of p53 in response to oxidative stress induced by mitochondrial respiration. It can also interact with p53 and DNA-PK, inhibiting p53 phosphorylation and activation [72].

Downregulation of ARTEMIS occurs in chronic myeloid leukaemia cell lines, which are characterized by increased levels of reactive oxygen species (ROS) that lead to DNA damage, including DSBs. In these cells, the low levels of the protein cause abnormal processing of DSBs with decreased stability of DNA-PK complexes at DNA ends [73]. Hypomorphic mutations, resulting in truncation of the last exon, have been described in patients with aggressive Epstein-Barr virus-associated B-cell lymphoma (Table 1). Although these patients did not show SCID, they showed low diversity in V(D)J junctions [74,75]. These findings were confirmed in mouse models [76].

The fact that cells deficient in ARTEMIS are more sensitive to radiation has been used as a therapeutic approach. A peptide inhibiting the interaction between ARTEMIS and DNA ligase IV, which is needed for its nuclease activity, has been used as a radiosensitizer that delays DNA repair and synergizes with irradiation to inhibit cell proliferation and induce cell cycle arrest and apoptosis [77].

### 2.4. EXD2

EXD2 (3′–5′ exonuclease domain-containing protein 2) has a conserved exonuclease domain with high sequence similarity to WRN (explained below). It only functions as an exonuclease when the protein oligomerizes and it can discriminate substrate (DNA or RNA) depending on the metal cofactors [78]. EXD2 localizes at the mitochondrial membrane, where it regulates mitochondrial translation [79], and the nucleus, where it promotes genome stability by acting on replication forks and DSB repair. EXD2 is recruited to replication forks upon replication stress to counteract fork reversal by suppressing the uncontrolled degradation of nascent DNA, allowing efficient fork restart [80]. This protection of the replicating fork is shared with BRCA1/2. Therefore, in the absence of both proteins the unprotected replication forks collapse, resulting in genome instability and compromised cell survival [80]. EXD2 is also essential for the repair of DSBs by HR. It interacts with the MRN (MRE11-RAD50-NBS1) complex to accelerate 3′ resections of double-stranded DNA (dsDNA), both short- and long-range [81]. Cells deficient in EXD2 show spontaneous chromosomal instability and are sensitive to DNA damage induced by anti-cancer agents such as IR and campthotecin [81], thus EXD2 is a good target for the development of a new anti-tumour treatment. So far, no studies have analysed the expression or the presence of mutations in human tumours.

### 2.5. EXO1

Exonuclease 1 (EXO1) is a member of the Rad2/XPG family, which contains DNA endonuclease, RNase H, and 5′–3′ exonuclease domains [82]. EXO1 (together with FEN1 and POLD) is essential for removing primers and for Okazaki fragment maturation during replication [11,83]. It is also involved in several DDR pathways such as MMR, where it is recruited by MutSα, MutSβ, and MutLα to degrade the newly synthesized DNA containing the replication error [11,83]; and HR, where it resects DNA in DSBs to allow RAD51 loading and strand exchange [84,85,86]. Upon DNA damage, EXO1 is involved in the recruitment of translesion synthesis (TLS) polymerases to sites of UV damage [87] and in the enlargement of single-stranded DNA (ssDNA) gaps to activate the ATR checkpoint by NER [85,88].

EXO1 has been associated with different types of tumours and its overexpression causes an increase in its DNA repair activity and genome instability. Overexpression of EXO1 occurs in prostate [89,90], breast [91,92,93], ovarian (cell lines) [94], lung [95], liver [96,97], bladder [98] and melanoma [99] cancer patients (Table 1). Moreover, mutations in the exonuclease domain resulting in loss of function, such as the A153V and N279S mutations, are found in colorectal and small intestine tumours [100]. These types of tumours also present the E109K mutation, which does not disrupt exonuclease activity, but, as it is localized in the PAR-binding motif, hinders its recruitment to DNA damage sites. In addition, several *EXO1* polymorphisms have been associated with a high risk of prostate [101], ovarian [102], lung [103,104,105], oral [106], liver [107], colon [108] and stomach [109] cancer, whereas other variants have shown protective roles in tissues like liver [110] and colon [111].

The effects of EXO1 inactivation (E109K mutation) [85] or deletion (*Exo1* knockout (KO)) [112] have been studied in mouse models (Table 2). Both mutant mice showed significantly reduced survival and accelerated tumorigenesis compared to wt mice. However, they showed differences in tumour spectrum. While *Exo1* KO predominantly develops lymphomas, mutated mice (*Exo1^E109K^*) develop sarcomas and adenomas. The different patterns of tumorigenesis can be attributed to the DSBR deficiency in mutated mice whereas in *Exo1* KO mice both the DSBR and MMR pathways are disrupted.

EXO1 activates the immune system in mice with an MLH1-deficient background through the activation of the cGAS-STING pathway [113]. Under normal circumstances, MutLα regulates the activity of EXO1 to generate the appropriate length of ssDNA. However, in the absence of this regulation, EXO1 induces excessive DNA degradation, producing unprotected ssDNA. These events lead to DNA breaks, chromosome abnormalities, and the release of nuclear DNA into the cytoplasm leading to cGAS-STING pathway activation and thus a type I IFN innate immune response. Therefore, it has been proposed that combining radiation and immunotherapy in MLH1-defective patients will be beneficial.

### 2.6. EXOG

EXOG (Exo/Endonuclease G) is a mitochondrial (mt) endo/5′–3′exonuclease with a preference for ssDNA [114,115]. It forms a complex with the mt repair proteins to remove the 5′-blocking oxidized residues of SSBs in the mt genome by BER. Therefore, depletion of EXOG induces persistent SSBs in the mtDNA, enhances ROS levels, and induces mt dysfunction, triggering the intrinsic apoptotic pathway [116]. This mechanism is especially important in tissues with elevated levels of oxidative agents such as human lung adenocarcinoma tumours, and with high levels of hydrogen sulphide (H_2_S)-producing enzymes. Elevated levels of H_2_S stimulate mtDNA repair through sulfhydration of EXOG, which increases its interaction with mt repair proteins to enhance DNA repair [117].

EXOG participates in mtDNA replication. In this process, RNase H1 removes all the RNA primers apart from two nucleotides that remain attached to the 5′end of the nascent DNA. EXOG removes this dinucleotide of the RNA/DNA hybrid duplex, maintaining mitochondrial genome integrity [118]. Since the identification of EXOG in 2008, only one report has associated EXOG with cancer: a missense mutation was found in a familiar case of appendiceal mucinous tumours, an extremely rare disease with uncertain genetic aetiology [119].

### 2.7. FAN1

FANCD2/FANCI-associated nuclease 1 (FAN1) is a 5′ flap structure-specific endonuclease and 5′–3′ exonuclease with broad substrate specificity [12]. It is essential to maintain chromosomal stability and resolve ICLs. Although its exact mechanism of action remains unclear, it is thought that FAN1 makes 2–6 nucleotide incisions at the sides flanking the ICLs, generating a suitable substrate for other nucleases and polymerases. In addition, it can participate in MMR, interacting with MutLα in the absence of EXO1, or cleave D-loop structures formed during HR. In response to replication stress, FAN1 also controls the progression of stalled replication forks, where it is recruited by Ub-FANCD2 (Fanconi anemia pathway) [12].

A deficiency of FAN1 in humans leads to chromosomal abnormalities (caused by failure of the replication fork) that can cause rare kidney and neurological diseases such as schizophrenia, epilepsy, and autism [12,120]. Although the loss of FAN1 function does not increase the burden of cancer [121], some mutations have been found in tumours, including mutations abolishing nuclease/exonuclease activity. For example, the p.M50R mutation occurs as a germline mutation in hereditary pancreatic cancer [122] and it also increases the risk of colorectal cancer [123] (Table 1). Additional germline mutations have been suggested to increase susceptibility to colorectal cancer [124] and primary hepatic mucoepidermoid carcinoma [125]. Moreover, mice defective in the nuclease domain (Table 2) develop carcinomas and lymphomas [126].

Loss of *FAN1* leads to sensitivity to crosslinking agents, especially in BRCA2-deficient cells [127]. Increased FAN1 expression in tumours refractory to treatment has been observed in breast and ovarian cancers [128]. Therefore, inhibition of FAN1 could be used to sensitize cancer cells to conventional chemotherapy. Additionally, FAN1 functional status in cancer cells might be used as a biomarker to predict response to treatment.

### 2.8. FEN1

Flap endonuclease 1 (FEN1), also known as DNase IV, belongs to the RAD2 family and is involved in multiple functions via different catalytic activities [13]. FEN (flap-specific endonuclease) activity is responsible for RNA primer removal in the maturation of Okazaki fragments during DNA replication and repairing DNA lesions that have an oxidatively damaged sugar moiety in a PCNA-dependent BER pathway called long-patch BER. EXO (5′ exonuclease) and GEN (gap-endonuclease) activities are important for the resolution of trinucleotide repeat sequence-derived DNA hairpin structures, oligonucleaosomal fragmentation of chromosomes in apoptotic cells, and the resolution of stalled replication forks caused by exogenous insults. In this case, FEN1 forms a complex with WRN to arrest the replication fork and resolve the chicken foot structure or cleave the fork to start the break-induced recombination. These multiple functions are regulated protein-protein interactions, post-translation modifications, and cellular compartmentalization, for example, FEN1 translocates to the nucleus upon DNA damage [13].

FEN1 somatic mutations have been found in non-small cell lung carcinoma, melanoma, and oesophageal cancers, some of them inactivating its exonuclease activity [129]. To study the role of FEN1 in cancer, mouse models have been developed (Table 2). *Fen1^+/−^* mice (*Fen1* KO is embryonically lethal [130]) have an increased risk of tumour development, especially lymphomas [131], and tumorigenesis is further increased in combination with other alterations such as *Apc^1638N^*; these mice present reduced survival and increased intestinal adenocarcinomas compared to *Apc^1638N^* alone [131]. Mice expressing the FEN1 E160D mutation (abrogates the EXO and GEN activities but not FEN activity), which leads to spontaneous mutations and the accumulation of incompletely digested DNA fragments in apoptotic cells [129], developed autoimmunity, chronic inflammation, and lung, testis/ovary, liver, kidney, spleen, stomach and lymphoma cancers. This phenotype is related to higher spontaneous mutation rates and the accumulation of apoptotic DNA in mutated cells leading to the DNA damage response and inflammation. Another example is the L209P mutation, found in colorectal cancer patients [132]. This mutated protein has lost all three activities and acts as a dominant-negative isoform. Mutated cells show high sensitivity to DNA damage, which causes genomic instability and transformation.

FEN1 is expressed in proliferating cells and is overexpressed in different tumours such as prostate [133], testis [134], lung [134,135], brain [134], gastric [136] and breast [137,138] (Table 1). In some cases, its overexpression is correlated with hypomethylation of the *FEN1* promoter and linked to increased tumour grade and aggressiveness [136,137]. *FEN1* polymorphisms have been associated with an elevated risk of lung, ovary, bladder, breast, glioma, and digestive cancers. In contrast, a protective role was attributed to some other variants in oesophagus, breast, and leukaemia cancers [139,140,141,142,143,144,145,146,147,148,149,150,151,152].

### 2.9. MRE11A

MRE11A is an ssDNA endonuclease/dsDNA 3′–5′ exonuclease of the MRN complex that is involved in DNA repair (HR and alternative NHEJ) following DSBs lesions, meiotic recombination, cell cycle checkpoints, and maintenance of telomeres. Its exonuclease activity plays an essential role in DDR, degrading DNA between the endonucleolytic incision sites, which creates an entry site for the long-range resection nucleases [153]. Mutations in MRE11A have been found in some types of cancer characterized by chromosomal instabilities such as breast, endometrium, and colon [154,155,156,157]. Mutations Y187C and H52S inactivate MRE11A exonuclease but not endonuclease activity [158]. Some frameshift mutations generate splicing variants that lead to exon loss. HCT116 cells (colon cancer cell line) have a mutant protein without exons 5–7, where the exonuclease domain is located, leading to the accumulation of unrepaired DNA [156]. MRE11A has also demonstrated potential as a predictive marker for radiotherapy in bladder cancer patients (Table 1), where high expression of MRE11A has been associated with a good prognosis [159].

### 2.10. p53

p53 is known to be the “guardian of the genome”, ensuring genetic stability through several roles that include control of the cell cycle, senescence, apoptosis, and DNA repair in response to oncogene activation, DNA damage, and other stress signals. Although p53 acts mainly as a transcriptional factor, it has 3′–5′ exonuclease activity in its core domain, where the sequence-specific DNA binding domain is located [160,161,162,163]. These two activities are mutually exclusive; therefore, exonuclease activity is mainly cytoplasmic [164,165,166]. It has been suggested that its exonuclease function is its most ancient function since this domain is present even in the ancestral p53 in invertebrates [167]. p53 shows a preference for ssDNA (although it can also process dsDNA, single-stranded RNA (ssRNA), and double-stranded RNA (dsRNA)), can remove 3′-terminal miss-pairs, and has a proofreading function when interacting with exonuclease-deficient polymerases [164,168,169]. Exonuclease activity has been observed in unstressed cells, but it also responds to exogenous stimuli such as DNA-damaging agents. In this scenario, exonuclease activity is not involved in cell cycle arrest but is essential for the induction of apoptosis in DNA-damaged cells [165]. Several core domain mutations have been found in cancer. However, their specific effect on exonuclease activity has not been assessed.

### 2.11. PLD3 and PLD4

PLD3 and PLD4 belong to the phospholipase D (PLD) family [170] and are characterized by their HKD motifs. They are glycosylated transmembrane proteins located in endolysosomes and surprisingly, in contrast to their family members PLD1/2, do not possess phospholipase activity. Instead, their different HKD amino acid composition allows them to process ssDNA from 5′ to 3′ [26], degrading endogenous ssDNA and thereby preventing autoimmune pathologies like rheumatoid arthritis [29]. In addition, both can degrade ssRNA, thus limiting autoinflammation triggered by both endosomal TLR and cytoplasmic STING nucleic acid sensing pathways [27].

PLD family members have a well-established role in promoting tumorigenesis in multiple types of cancers [171]. Although PLD4 and PLD3 have been linked through murine models (Table 2) and genome-wide association studies to autoinflammatory diseases [172] and late-onset Alzheimer’s disease [173], respectively, very little evidence exists regarding their involvement in cancer. PLD3 has been associated with a favourable prognosis in osteosarcoma and included in a prognostic gene signature of immune cell infiltration [174,175] (Table 1). Likewise, PLD4 expression has been proposed to predict, together with other genes, better survival of HER2-positive breast cancer patients [176] and, when expressed in M1 macrophages, it might have antitumoral effects in colon cancer [177]. Overall, our knowledge of the role of PLD3/4 in cancer is limited, and thus additional research is warranted.

### 2.12. POLD and POLE

Polymerase delta (POLD) and polymerase epsilon (POLE) contain 3′–5′ exonuclease proofreading domains. Several studies have suggested that this intrinsic proofreading exonuclease activity plays a critical role in suppressing carcinogenesis. Increased epithelial cancer was observed in mice deficient for *Pold* proofreading exonuclease (Table 2) (*Pold1^D400A/D400A^*) [178]. Moreover, germline or somatic mutations in the exonuclease domain of POLE were found to increase the mutation rate and risk of cancer development in the colon and endometrium [179]. Although somatic *POLD* exonuclease domain mutations are less frequent, they were observed in colon, endometrium, and melanoma cancers [180,181] (Table 1). Patients with these mutations have an excellent prognosis and respond well to immunotherapy because the high mutation rate increases the probability of neoantigens, which are recognized by the immune system [179,181].

### 2.13. RAD9A

RAD9A is a multifunctional protein that contains a 3′–5′ exonuclease domain in its N-terminal portion [182], although the exact function of this activity remains unknown. This protein is involved in several cellular functions such as apoptosis (it contains a BH3 domain), but its main role is to control the cell cycle checkpoint and DNA damage repair as an early DNA damage sensor of SSBs and DSBs [182]. RAD9A participates in multiple repair pathways such as BER (interacting and activating APE1), MMR (interacting with MLH1, MSH2, MSH3, and MSH6), ICL (activating FANCD2 through ATR activation), and HR (interacting with RAD51) [182]. Consequently, cells deficient in RAD9A show spontaneous chromosome aberrations and are more sensitive to DNA-damaging agents such as hydroxyurea, UV light, and IR [183].

Owing to its involvement in multiple and varied cellular functions, RAD9A has a dual role in cancer, acting as a tumour suppressor or promoter depending on the tissue. RAD9A is overexpressed and accumulated in the nucleus of samples from non-small cell lung carcinomas, and is correlated with high proliferation [184,185]. Overexpression of *RAD9* was also observed in thyroid [186], prostate [187,188], and breast cancer [189] (Table 1). In breast and prostate cancer, overexpression is due to amplification of the 11q13 chromosome (where the *RAD9* gene is located) or differential intron methylation in the *RAD9* gene. The introns contain sequences that inhibit *RAD9* expression but are suppressed upon methylation, an event that occurs in childhood leukaemia [190]. Increased levels of RAD9A were correlated with bigger tumours, local recurrence, and higher aggressiveness. However, other types of tumours like gastric carcinomas showed decreased expression [191]. The same could be true for skin cancers, since the skin conditional *Rad9*-deficient mouse (total KO is lethal [192]) showed enhanced tumour development upon application of carcinogen [193] (Table 2).

### 2.14. TREX1

Three-prime repair exonuclease 1 (TREX1 or DNase III) is a non-processive 3′–5′ exonuclease that degrades ssDNA and dsDNA from the 3′-ends [194,195]. TREX1 is ubiquitously expressed and localized in the endoplasmic reticulum membrane. It plays a major role in DNA-driven immune responses, where it is involved in self- and non-self-DNA degradation, limiting the activation of DNA-sensing and -signalling pathways, such as the cGAS-STING pathway. Thus, it prevents aberrant interferon-mediated inflammatory responses and autoimmunity [196,197,198,199,200]. Furthermore, TREX1 has been implicated in DNA degradation during granzyme A-mediated cell death [21].

Independently of its exonuclease activity, TREX1 reduces glycan-driven immune responses by interacting with the oligosaccharyltransferase complex, contributing to its correct function [201]. Consistently, loss of function mutations have been associated with inflammatory and autoimmune diseases. Of note, mutations located in the exonuclease domains (N-terminus region) are mostly linked to Aicardi-Goutières syndrome and systemic lupus erythematosus. Mutations in the endoplasmic reticulum localization and oligosaccharyltransferase interaction domain (C-terminus region) are mainly linked to retinal vasculopathy with cerebral leukodystrophy [202,203,204,205]. In agreement with the major role of TREX1 as an anti-inflammatory player, *Trex1* knockout mice [206] and TREX1^D18N^ exonuclease defective mice [207,208] develop an inflammatory systemic phenotype, but they are not tumour prone (Table 2). Nevertheless, TREX1 can influence genomic stability and DDR in distinct ways. For instance, TREX1 is induced after DNA damage, favouring DNA repair [19] and interacting and stabilizing PARP1 [209]. Moreover, TREX1-deficient cells exhibited increased levels of p53 and p21 and ATM-dependent checkpoint activation [210], which suggests chronic activation of DDR. However, TREX1 can drive chromosome mis-segregation and error-prone DNA repair in tumoral cells undergoing telomere crisis, thus fostering genomic instability [211]. Finally, TREX1 locates to micronuclei, degrading DNA when their membranes break, preventing its cytoplasmic sensing and supporting the chromosomal instability of tumours [197].

Depending on the type of tumour, TREX1 expression is upregulated or downregulated (Table 1). TREX1 overexpression is observed in oesophageal [212] and cervical [213] cancers but is downregulated in melanoma [214] and osteosarcoma [215], where TREX1 expression is only observed in non-metastatic patients. Conflicting results have been found in breast cancer, with some studies showing overexpression and some downregulation [216,217]. Recently, a potentially pathogenic TREX1 variant was found in a small cohort of familial colorectal cancer type X (FCCTX), although the functional consequences of the variant were not assessed [218]. Of note, TREX1 is induced in carcinoma cells by irradiation [199,219] and in glioma [214], melanoma [220], and nasopharyngeal [221] cells by anticancer drugs, triggering a pro-survival response. Because preventing TREX1 from degrading accumulated cytosolic DNA renders the cGAS-STING pathway active with the consequent production of type I IFNs, TREX1 has attracted great interest as a target to elicit antitumour immunity [222].

### 2.15. TREX2

Three-prime repair exonuclease 2 (TREX2) is a 3′–5′ exonuclease that is highly homologous to TREX1 [223,224], sharing similar biochemical and structural features [225]. However, in contrast to TREX1, TREX2 shows restricted expression in keratinocytes, localizes in nuclei, and plays a relevant pro-inflammatory role in keratinocytes [22,224]^,^ [226] without activating DNA-driven immune responses [227]. TREX2 facilitates nuclear DNA degradation in stressed keratinocytes, thus promoting cell death [22,224,226,227]. Interestingly, TREX2 has been shown to improve targeted CRISPR/Cas9 efficiency [228,229,230] by increasing deletion size and preventing perfect DNA repair, thereby avoiding repeated Cas9 cleavage and chromosomal translocations [231].

Studies on the role of TREX2 as a tumour suppressor have reported contrasting results. In artificial settings, chromosomal and genomic instability were reported using *Trex2*-null and mutated embryonic stem cells [232,233]; however, TREX2 deficiency in mice (Table 2) does not lead to a tumour prone phenotype [22,224]. Depending on the DNA repair status of embryonic stem cells, TREX2 may display a dual function in the DDR pathway, dependent and independent of its exonuclease activity, facilitating or avoiding replication fork instability and mutations [234,235]. However, neither genomic nor chromosomal instability are observed in cells from *Trex2* knockout mice or in keratinocytes, where *Trex*2 is highly expressed, nor in embryonic stem cells, in which TREX 2 expression is not detectable [224]. Nevertheless, *Trex2* knockout mice show increased susceptibility to DNA damage-induced skin tumorigenesis. TREX2 interacts with phosphorylated H2AX histone, which is a critical player in both DNA repair and cell death and is recruited to low-density nuclear chromatin and micronuclei. Upon DNA damage, TREX2 participates in DNA repair but mostly contributes to DNA degradation of fragmented DNA, promoting cell death of damaged keratinocytes and favouring an antitumoral immune response, supporting a major role of TREX2 as a proapoptotic tumour suppressor in keratinocyte-driven tumours [22,224].

TREX2 deregulation and genetic alterations in cancer mostly indicate the role of TREX2 as a tumour suppressor. In cutaneous squamous cell carcinomas (cSCCs) and head and neck SCCs (HNSCCs) high expression of TREX2 was found in well-differentiated tumours while its expression was not detected in metastatic samples [22] (Table 1). Moreover, epigenetic regulation of TREX2 was observed in colorectal and laryngeal cancer [236]. Reduced DNA methylation in the TREX2 intragenic gene locus is associated with elevated expression and better overall survival of patients. In contrast, TREX2 missense mutations and upregulation in colorectal cancer have been associated with reduced survival [237]. In this regard, SNPs in *TREX2* are more frequent in patients with head and neck SCCs than in healthy individuals [22].

### 2.16. WRN

Werner Syndrome protein (WRN) (also known as RECQL2) is a multifunctional protein that contains four functional domains, including a 3′–5′ exonuclease domain that can recognize a variety of DNA substrates [238]. This protein is crucial for genome stability through its involvement in DNA replication, recombination, and repair, although the specific relevance of the exonuclease domain has not yet been determined. Germline mutations in WRN lead to Werner Syndrome, characterized by premature aging and higher susceptibility to a broad spectrum of epithelial and mesenchymal tumours like sarcomas, melanoma, thyroid, and breast cancer [238]. Although no somatic mutations have been described in sporadic tumours, its expression is downregulated due to epigenetic inactivation or loss of heterozygosity in several solid tumours such as colorectal cancer and breast tumours [239,240]. Low expression of nuclear WRN has been associated with a worse prognosis and promoter hypermethylation as a predictor of good clinical response to DNA-damaging treatments [239,240,241,242].

**Table 1 cells-11-02157-t001:** Exonuclease alterations in cancer.

Gene	Alteration	Type of Cancer	Biomarker	Ref.
**AEN**	High expression	Prostate	High-risk recurrence	[40]
		Colorectal	Reduced survival	[41]
**APE1**	Exonuclease mutations	Glioblastoma, endometrial		[59,60]
	High expression	Lung, colorectal, cervical, prostate, bladder, gastric, hepatic, glioblastoma, osteosarcoma, head, and neck, ovarian, breast	Tumour aggressiveness, poor prognosis	[55,56]
**ARTEMIS**	Hypomorphic mutations	Lymphoma	High risk	[74,75]
**EXO1**	Exonuclease inactivating mutations	Colorectal tumours, small intestine tumours		[100]
	High expression	Prostate, breast, lung, liver, bladder, melanoma		[89,90,91,92,93,95,96,97,98,99]
**FAN1**	Exonuclease inactivating mutations	Pancreatic, colorectal, hepatic	High risk	[122,123,124,125]
	High expression	Ovarian	Poor prognosis	[128]
**FEN1**	High expression	Prostate, testis, lung, brain, gastric, breast	Increased tumour grade and aggressiveness	[133,134,135,136,137,138]
	SNP	Lung, ovary, bladder, breast, glioma, digestive	High risk	[139,140,141,142,143,144,145,146,147,148,149,150,151,152]
		Esophagus, breast, leukemia	Protective role	[144,146,147]
**MRE11A**	Exonuclease inactivating mutations	Breast, endometrium, colon		[154,155,156,157]
	High expression	Bladder	Good prognosis	[159]
**PLD3**	High expression	Osteosarcoma	Good prognosis	[174,175]
**PLD4**	High expression	HER2-positive breast cancer	Better survival	[176]
**POLD**	Somatic exonuclease domain mutations	Colon, endometrium, and melanoma	Good prognosis	[179,180,181]
**POLE**	Exo domains mutated	Colon, endometrium	High risk and increased mutation rate	[179]
**RAD9**	High expression	Lung, thyroid, prostate, breast	Bigger tumours, recurrence, and aggressiveness	[184,185,186,187,188,189]
	Low expression	Gastric		[191]
**TREX1**	High expression	Esophageal, cervix		[212,213]
	Low expression	Melanoma, osteosarcoma		[214,215]
**TREX2**	High expression	Low-grade HNSCC, laryngeal	Good prognosis	[22,236]
		Colorectal	Reduced survival	[237]
	Low expression	Metastatic HNSCC		[22]
**WRN**	Somatic mutations	Sarcomas, melanoma, thyroid, breast		[238]
	Low expression	Colorectal, breast	Bad prognosis	[239,240]

**Table 2 cells-11-02157-t002:** Murine strains modelling exonuclease gene functions.

Exonucl.	Mutant Mice	Alteration	Phenotype	Ref.
**APE1**	*Ape1^−/−^*	Gene deletion	Lethal	[50]
	*Ape1^+/−^*	Hemizygous	Cancer prone, lymphomas, sarcomas & adenocarcinomas	[52,53,54]
**ARTEMIS**	*Art^N/N^*	Gene deletion	Severe combined immunodeficiency	[70]
	*Art^N/N^Trp53^N/N^*	Gene deletion	Increased carcinogenesis in Art vs. p53 null mice	[72]
**EXO1**	*Exo^−/−^*	Gene deletion	Lymphoma; reduced survival; sterility	[112]
**FAN1**	*Fan^nd/nd^*	Nuclease defective	Cancer prone, carcinomas & lymphomas	[126]
**FEN1**	*Fen1^−/−^*	Gene deletion	Lethal	[130]
	*Fen1^+/−^*	Hemizygous	Tumours, mainly lymphomas	[131]
	*Fen1^+/−^ Apc^1368N^*	Hemizygous Mutation	Increased adenocarcinomas & decreased survival compared to Apc^1268N^	[131]
	*Fen1^E160D^*	Inactivation of exo- & endonuclease activities	Autoimmunity, chronic inflammation, and tumours. Spontaneous mutations; accumulation of non-digested DNA in apoptotic cells.	[129]
**PLD3 and PLD4**	*Pld3^−/−^*	Gene deletion	No inflammation	[26]
	*Pld4^−/−^*	Gene deletion	Inflammation, splenomegaly, high IFNγ levels	[26]
	*Pld3^−/−^Pld4^−/−^*	Gene deletion	Lethal liver inflammation, hemophagocytic lymphohistiocytosis, high IFNγ levels	[27]
**POLD**	*Pold1^D400A^*	Exonuclease domain mutated	Increased epithelial cancer	[178]
**RAD9**	*Rad9^−/−^*	Gene deletion	Lethal	[192]
	*Rad9^K5−/−^*	Gene deletion in keratinocytes	Enhanced tumour development upon exposure to carcinogen	[193]
**TREX1**	*Trex1^−/−^*	Gene deletion	Not cancer-prone. Inflammatory myocarditis	[206]
	*Trex1^D18N^*	Exonuclease defective	Not cancer-prone. Systemic inflammation. Lupus-like inflammatory syndrome.	[208]
**TREX2**	*Trex2^−/−^*	Gene deletion	Not cancer-prone. Increased carcinogenesis upon exposure to genotoxins. Reduced inflammation.	[22,224,226]

## 3. Outlook

The DDR encompasses different sensor and effector proteins, including exonucleases, that work together with the final aim of limiting damage, both for the cell and the organism. Almost all living organisms have evolved to possess a battery of mechanisms to ensure the preservation of their hereditary material, demonstrating the ubiquitous urgency to respond to damaged DNA [243,244]. DNA exonuclease activity is required for basic cell processes, such as the synthesis and repair of damaged DNA, cell death, and inflammation (Figure 1) which are important to maintain homeostasis and prevent diseases and cancer. While not all DNA exonucleases are directly involved in the DDR, their actions can indirectly alter it. The functional interplay between the exonucleases described in this revision is surprisingly significant (Figure 2) considering that some of them are highly confined and spatially separated in different tissues and compartments of the cell. While most exonucleases reside in the nucleus, EXOG and EXD2 are specifically located in the mitochondria, and TREX1, PLD3, and PLD4 in the cytosol. Additionally, others have fluctuating levels depending on the cell cycle (APEX1) or are only expressed in keratinocytes (TREX2). Either interacting in multiprotein complexes or by their own action, exonucleases contribute in different manners to execute a proper DDR.

DNA exonucleases might be required for the survival of some cancer cells that acquire dependency on normal cellular functions, such as DNA repair, and thus, produce a non-oncogene addiction [245]. DNA exonucleases may favour the appearance of mutations in healthy cells, converting them into tumour cells, or may be necessary for certain tumours to keep the inherent genomic instability of malignancy under control, aiding tumour cells to avoid cell death, or impact on the immune response. In this manner, aberrant DNAse activity in tumours have been suggested to be exploited as a molecular whistleblower for diagnosis [246]. Hence, these types of enzymes are promising targets for cancer treatment promoting synthetic lethality and early detection of certain types of cancer. For instance, blocking the function of those exonucleases that degrade cytoplasmic DNA preventing inflammation, such as TREX1, PLD3 and PLD4, would activate DNA sensors producing an IFN response and leading to antitumor immunity.

Exonuclease genetic alterations, changes in their activity and abnormal expression in human tumours together with functional studies with murine models point to a clear contribution of these proteins to cancer onset and development (Table 1 and Table 2). It is puzzling how, depending on the type of cancer, some exonucleases are overexpressed or downregulated. It is worth noting the different tissue-specific metabolic requirements and byproducts as well as the differences in protein expression that might influence the types of genes being expressed. Moreover, some types of cancers are characterized by a particularly elevated genomic instability and this itself could influence the expression of the transcription factors required for exonuclease expression. Nonetheless, each individual tumour is different and the genetic background of the individual as well as environmental factors such as diet, temperature, medications, or immune cell infiltration can influence gene expression, and result in these not surprising differences. On the other hand, the existing knowledge on some exonucleases, such as AEN, EXD2, EXOG, PDL3/4, TREX1, and TREX2, is relatively poor and sometimes conflicting. Advance in the understanding of the mechanisms and functions of each specific exonucleases would strengthen their value as potential targets for cancer treatment and/or as biomarkers.

## Figures and Tables

**Figure 1 cells-11-02157-f001:**
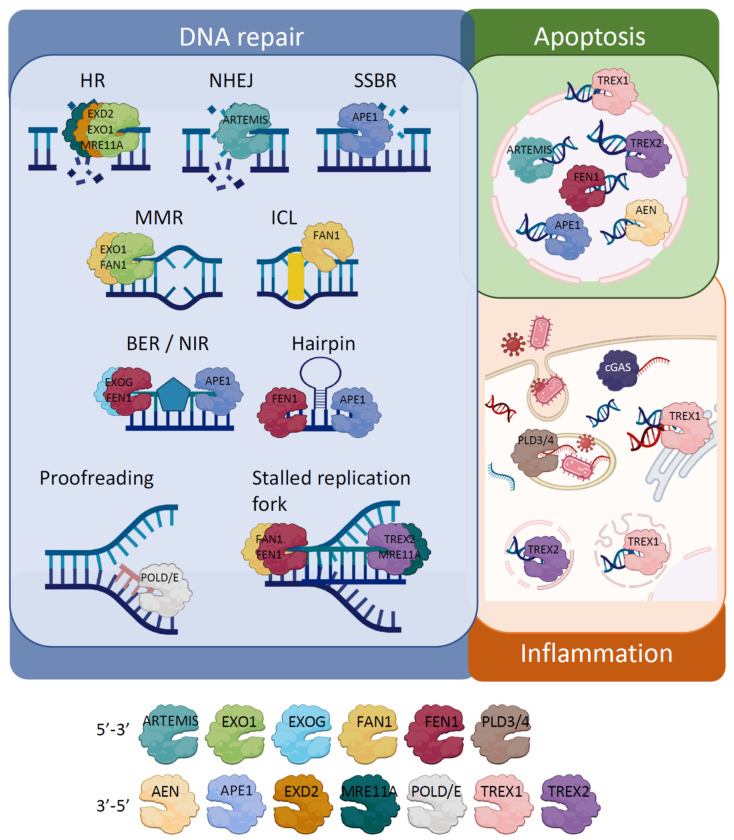
Exonucleases in DNA damage repair, apoptosis, and inflammation. Key exonuclease proteins in DNA damage repair, apoptosis, and inflammation processes are depicted. Exonuclease activity (5′–3′ and 3′–5′) is shown.

**Figure 2 cells-11-02157-f002:**
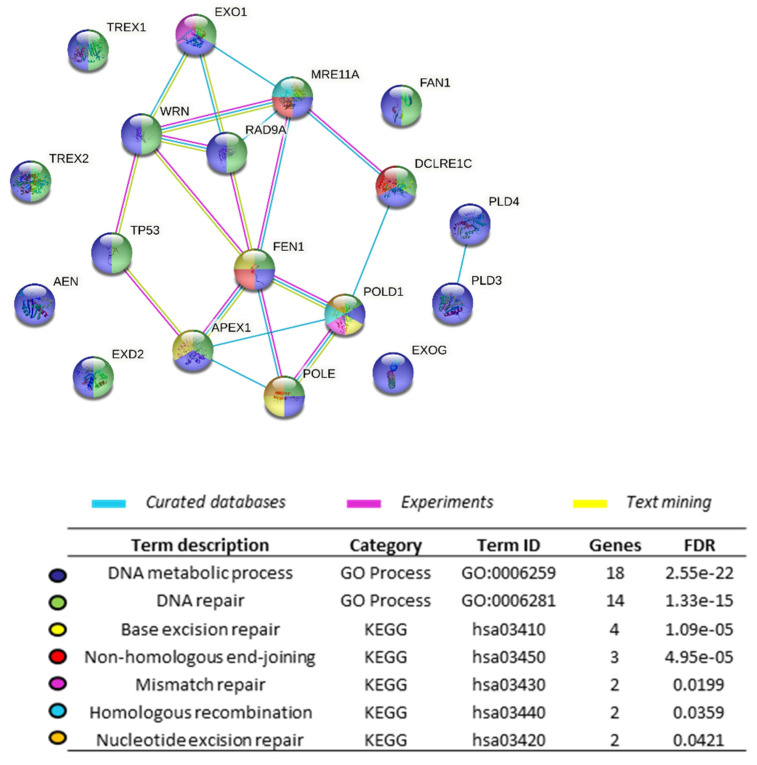
Exonuclease protein–protein interaction map. The interaction network of selected exonucleases was generated with STRING database v11.5 using basic settings selecting a physical subnetwork (the edges indicate that the proteins are part of a physical complex, although they may not directly interact) and medium confidence of 0.4. Proteins are represented as nodes, and lines indicate associations based on known functional interactions in humans. The network is significantly enriched in interactions (PPI enrichment *p*-value: 1.07e–13, FDR  <  0.05). All the proteins are included in the GO-term DNA metabolic process (in blue) (GO:0006259).

## Data Availability

Figure 1 was created with BioRender.com. The functional association between exonucleases was assessed and graphically represented using String: https://string-db.org/.

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
