# Peer review of "Exonucleases: Degrading DNA to Deal with Genome Damage, Cell Death, Inflammation and Cancer"

_cells, 2022, doi:10.3390/cells11142157_

Round 1

Reviewer 1 Report

 As described by the authors, exonucleases play paramount important roles in central pathways of the cell biology, e.g., DNA replication, repair, and death, as well as tuning the immune response. In this review, the authors provide latest summary on the identified Endo-/exonucleases, the impact of DNA degradation in DNA damage response and their links to inflammation and cancer. The review is comprehensive and very informative, and also interesting. The authors also provide their own insights into the exonucleases and DNA degradation.  The review can be accepted for publication.

Author Response

The English language and style have been revised by a professional editing service.

Reviewer 2 Report

Dear Authors, your manuscript is quite focused. The work is important and highlights DDR enzymes well. 

I would advise adding the larger landscape of the role of exonucleases added to the discussion to widen it. The possible interactions of the enzymes can also be discussed in the last section. I also advise that the language of the manuscript is improved.

Reviewer 3 Report

In this review Manils et al. present a great overview on exonucleases and the current knowledge on their involvement in cancer. The manuscript is clearly structured and very comprehensive.

Basically, I have no major concerns, but it would be nice if the authors could provide more synthesis. At this point the review is only a statement of facts. I am missing interpretation and the authors view on the data, especially in cases where there are contradictory data in different tumor types. As the review is on the role of exonucleases in cancer maybe the authors could change the title to reflect that. It would also be nice if the authors could provide a table summarizing the role of the described exonucleases in human cancer indicating whether they are tumor promoting or suppressing and whether there are inhibitors available (or in clinical trials?) and which exonucleases would have potential and why as targets of anti-cancer therapeutics.

Round 2

Reviewer 3 Report

I thank the authors for the thorough revision of their manuscript. The authors addressed all my points and present a significantly improved manuscript.